# Mobilization of Endogenous CD34+/CD133+ Endothelial Progenitor Cells by Enhanced External Counter Pulsation for Treatment of Refractory Angina

**DOI:** 10.3390/ijms251810030

**Published:** 2024-09-18

**Authors:** Joseph T. Tartaglia, Carol A. Eisenberg, Joseph C. DeMarco, Gregory Puccio, Christina E. Tartaglia, Carl V. Hamby

**Affiliations:** 1Department of Medicine, New York Medical College, Valhalla, NY 10595, USA; josephtartagliamdpc@gmail.com (J.T.T.); carol_eisenberg@nymc.edu (C.A.E.); 2Summit Health, Florham Park, NJ 07932, USA; 3Family Dentistry, Parkville, MD 21234, USA; 4Precision AQ, New York, NY 10165, USA; 5Department of Pathology, Microbiology, and Immunology, New York Medical College, Valhalla, NY 10595, USA

**Keywords:** adult stem cells, endothelial progenitor cells (EPCs), endothelial-to-hematopoietic stem cell (HSC) transition, enhanced external counterpulsation (EECP), refractory angina (RA), ischemic coronary artery disease

## Abstract

Adult stem cell therapy via intramyocardial injection of autologous CD34+ stem cells has been shown to improve exercise capacity and reduce angina frequency and mortality in patients with refractory angina (RA). However, the cost of such therapy is a limitation to its adoption in clinical practice. Our goal was to determine whether the less costly, less invasive, and widely accessible, FDA-approved alternative treatment for RA patients, known as enhanced external counterpulsation (EECP), mobilizes endogenous CD34+ stem cells and whether such mobilization is associated with the clinical benefits seen with intramyocardial injection. We monitored changes in circulating levels of CD34+/CD133+ and CD34+/KDR+ cells in RA patients undergoing EECP therapy and in a comparator cohort of RA patients undergoing an exercise regimen known as cardiac rehabilitation. Changes in exercise capacity in both cohorts were monitored by measuring treadmill times (TT), double product (DP) scores, and Canadian Cardiovascular Society (CCS) angina scores between pre- and post-treatment treadmill stress tests. Circulating levels of CD34+/CD133+ cells increased in patients undergoing EECP and were significant (β = −2.38, *p* = 0.012) predictors of improved exercise capacity in these patients. CD34+/CD133+ cells isolated from RA patients could differentiate into endothelial cells, and their numbers increased during EECP therapy. Our results support the hypothesis that mobilized CD34+/CD133+ cells repair vascular damage and increase collateral circulation in RA patients. They further support clinical interventions that can mobilize adult CD34+ stem cells as therapy for patients with RA and other vascular diseases.

## 1. Introduction

Refractory angina (RA) is defined as the chronic occurrence (greater than or equal to 3 months) of frequent angina attacks uncontrolled by a combination of optimal drug treatment, coronary angioplasty, and coronary bypass surgery and where reversible coronary ischemia has been clinically determined to be the cause. The increasing number of survivors from revascularization procedures in an aging population is increasing the prevalence of RA [1]. It is estimated that perhaps due to increasing survival age, there are increasing numbers of cases, up to 75,000 new cases of RA diagnosed yearly in the United States [2]. Since the underlying cause for RA is thought to be insufficient coronary artery circulation, one avenue of research that has been explored for the treatment of these patients is the injection of autologous adult CD34+ stem cells under the assumption that such cells, with the demonstrated capacity to differentiate into endothelial cells, can enter and repair damaged coronary vasculature [3]. In support of this approach, a meta-analysis of three clinical trials conducted between 2007 and 2016 demonstrated that intramyocardial injection of autologous CD34+ adult stem cells improved exercise capacity, reduced angina frequency, and reduced mortality in RA patients [4]. Four additional clinical trials conducted between 2016 and 2022 have shown that intramyocardial or intracoronary injection of CD34+ cells improved clinical outcomes for RA patients, as reviewed by Hassanpour et al. [3], who note that the FDA (Food and Drug Administration) has designated CD34+ cell products as “regenerative medicine advanced therapy” for refractory angina [3]. However, this therapy’s application has been limited by the cost of the procedures and the need for validation from additional clinical studies. In this investigation, we addressed the question of whether an alternative therapy called enhanced external counterpulsation (EECP) that is a noninvasive, FDA-approved therapy for angina refractory to standard treatments may achieve its clinical benefits through mobilization of particular subpopulations of CD34+ stem cells [5]. The procedure consists of external inflatable cuffs that wrap around the lower legs, upper legs, and buttocks. These computer-controlled cuffs increase arterial blood return to the coronary arteries as they inflate in diastole. The cuffs then deflate in systole with each heartbeat, decreasing systemic resistance and acting to increase cardiac output. The combined action subjects the coronary vasculature to higher blood flow than can be achieved with exercise alone. Since the first randomized, double-blinded, placebo-controlled trial in the U.S., the MUST EECP trial was conducted, showing EECP can reduce angina and extend treadmill times to exercise-induced ischemia [6]. EECP currently has an IIb indication by the United States Food and Drug Administration for the management of RA. Registry data of EECP have verified the clinical benefits for the majority of the patients treated [7,8,9]; however, the long-term clinical efficacy remains equivocal [10,11]. It has been proposed that stem cells released into the circulation in patients being treated with EECP participate in revascularization and/or repair of damaged arteries [12,13] and increase collateral circulation in ischemic areas [14]. Support for this mechanism comes from two groups who have reported independently that patients treated with EECP experienced increases in CD34+ endothelial progenitor cell (EPC) stem cell subpopulations in their peripheral blood [15,16] while animal and human studies [9,10,11,17,18,19,20,21,22] and direct evidence in human patients treated with EECP for stable angina have demonstrated an increased collateral flow index [23]. Over the years, support for this hypothesis has come from early clinical studies which demonstrated that the percentage of patients who have improved myocardial perfusion scans after completing EECP treatment is greater in patients who are able to exercise to higher workloads on a treadmill than those who are not [24,25]. These findings indicate that (1) EECP promotes the development of collateral circulation in ischemic regions in patients who respond to treatment with higher cardiac workloads, and (2) the increase in collateral circulation is not solely due to an exercise training effect on the skeletal muscles. We anticipate that patients who mobilize adult EPCs in response to EECP will achieve higher cardiac workloads through improved collateral circulation and will therefore experience the greatest long-term clinical benefits.

This study was designed to test the hypotheses that (1) endogenous adult CD34+/CD133+ and CD34+/KDR+ EPCs are mobilized in RA patients during EECP and (2) that increases in these cell populations positively correlate with improved exercise capacity in subjects as measured by cardiac stress tests. Therefore, we measured the peripheral blood levels of three stem cell populations: total CD45+/CD34+stem cells, CD34+/CD133+ cells, and CD34+/KDR+ cells in subjects prior to, during, and at the conclusion of treatment with EECP and alternative exercise therapy to compare changes in EPC levels to objective measures of exercise capacity as assessed by pre- and post-treatment exercise stress tests. In addition, patients were followed clinically to determine whether these interventions reduced the hazard of MACE). The STROBE 2007 checklist for clinical cohort studies is provided in Appendix A.

## 2. Results

### 2.1. Subject Characteristics

Thirty-nine consecutive patients with ischemic coronary artery disease were studied prospectively for EECP-induced changes in exercise capacity to determine if they were associated with changes in EPC levels during treatment or with MACE-free survival following treatment. A pre-therapy stress test was not performed on one patient, and post-therapy tests were not performed on four patients due to the occurrence of MACE in two cases and non-cardiac-related reasons in two other cases. Eleven patients who were candidates for EECP but elected to undergo physician-monitored cardiac rehabilitation exercise instead were enrolled as a separate comparator cohort. They were subjected to pre- and post-therapy stress testing like EECP patients and had blood samples drawn for EPC enumeration at the beginning, middle, and end of their therapy. The median clinical follow-up of all patients was 54 months, with a range of 5 to 112 months. None of the baseline clinical characteristics of the EECP cohort, which included age, body mass index, low-density lipid (LDL) levels, and high-density lipid (HDL) levels; a history of diabetes, hypertension, prior coronary artery bypass surgery, or smoking; or pharmacotherapy with antithrombotic agents, calcium channel blockers, diuretics, nitrate, pentoxifylline, or clopidogrel, were significantly associated with MACE risk (Appendix A). All but four EECP patients were male, all but three were taking aspirin, all but two were taking statins, and only one was not taking a beta blocker; thus, the statistical power for detecting an effect on MACE hazard for these variables is too low to be instructive. Therefore, they were not included in the survival analysis.

### 2.2. EECP-Induced Improvement in Exercise Capacity Is Associated with Mobilization of CD34+/CD133+ Stem Cells

Following EECP therapy, treadmill times (TT) and double product (DP) scores were improved in 20/37 (54%) and 13/37 (35%) of patients, respectively, and 8/11 (73%) cardiac rehabilitation patients had improved TT and 6/11 (55%) had improved DP scores (Figure 1). Seventy percent of EECP patients had at least a 1 class improvement in CCS score, and median post-EECP angina scores (CCS = 2) were significantly improved (*p <* 0.0001) over pre-EECP scores (CCS = 3). Forty-five percent of cardiac rehabilitation patients had at least a 1 class improvement in CCS score, but median post-rehab angina scores (CCS = 1) were not significantly improved (*p >* 0.200) over pre-rehabilitation scores (CCS = 2). Since there was no complete concordance between up or down changes in TT and DP scores in individual patients (Figure 1), exercise capacity was classified as improved when at least one treadmill parameter significantly increased while the other parameter was either also increased or not significantly reduced. When categorized in this way, 54% of patients who underwent EECP therapy responded with improved exercise capacity, while 73% of cardiac rehabilitation therapy had improved exercise capacity. Pre-treatment TT and DP scores were compared between responder and non-responder EECP patients and cardiac rehabilitation patients to see what, if any, differences in baseline measures of exercise capacity existed between them prior to initiation of therapy. Pre-treatment DP scores for EECP responders and non-responders were 16,735 ± 1051 and 19,561 ± 966 (*p* = 0.059), respectively, and were 18,244 ± 835 and 18,930 ± 3763 (*p* = 0.630), respectively, for rehabilitation patients. Similarly, baseline TT scores were not significantly different between responders and non-responders in EECP patients (317 ± 32 vs. 367 ± 27 s, *p* = 0.252) or rehabilitation patients (433 ± 31 vs. 345 ± 85 s, *p* = 0.376).

There were sufficient cell count data from 26 EECP patients for analysis of treatment-related changes in circulating EPCs. Their cell count data were compared to those of 11 cardiac rehabilitation patients. Differences in baseline counts of EPC subsets between groups were analyzed with the Kruskal–Wallis multiple comparison Z test due to non-normal distributions and unequal variances of the data. Median baseline CD34+/CD133+ cell counts per 100,000 MNCs were significantly greater in EECP patients (41, IQR = 66) than in rehabilitation patients (6, IQR = 17) while median baseline CD34+/KDR+ cell counts (39, IQR = 35 vs. 49, IQR = 98) and CD45+/CD34+ cell counts (258, IQR = 1109 vs. 145, IQR = 601) were not significantly different between the two groups. Longitudinal cell count data acquired during treatments were transformed to log_10_ values to obtain variance stabilized, normal distributions and circulating levels of CD34+/CD133+, CD34+/KDR+, and total CD45+/CD34+ cells were compared for changes between the first half (weeks 0–3) and second half (weeks 4–7) of EECP and start (baseline) and second half (middle + end) of cardiac rehabilitation. Circulating CD34+/CD133+ cell counts increased significantly between the first and second half of EECP treatment (*p* = 0.004), but neither CD34+/KDR+ nor CD45+/CD34+ cell counts increased significantly between the first and second half of treatment when averaged over all patients (Table 1). None of the CD34+ subsets were significantly increased by cardiac rehabilitation therapy when averaged over all patients. On average, EECP patients had significantly (all *p* < 0.003) higher numbers of circulating CD34+/CD133+, CD34+/KDR+, and CD45+/CD34+ cells both at the beginning and end of therapy than cardiac rehabilitation patients.

To determine if improved exercise capacity was associated with increases in circulating EPCs, the responder and non-responder patients for both treatments were analyzed separately for changes in EPC levels. The average level of CD34+/CD133+ counts rose significantly (*p* = 0.002) in the second half of EECP treatment for responders but not for non-responders (*p* = 0.996), whereas average counts of CD34+/KDR+ and CD45+/CD34+ cells did not change significantly (all *p* > 0.200) between the first and second half of treatment for either group (Figure 2). In contrast, there were no significant (all *p* > 0.35) treatment-related differences in the total CD45+/CD34+ stem cell and EPC populations when examining responder and non-responder cardiac rehabilitation patients (Figure 2). When analyzed by logistic regression for effects on exercise capacity, increased CD34+/CD133+ cell counts were found to be significant (regression co-efficient β = −2.38, *p* = 0.012) predictors of improved exercise in EECP patients but not in cardiac rehabilitation patients (*p* = 0.982). In contrast, increased CD34+/KDR+ cell counts were not significant (all *p* > 0.25) predictors of improved exercise in either EECP or cardiac rehabilitation patients.

The foregoing analysis of average cell counts of EPC subsets in the responder and non-responder groups does not capture treatment-associated changes that occurred in individual patients. Therefore, the percent change in log_10_ EPC cell counts over the course of treatment was calculated for each patient and compared between the responder and non-responder groups. These data were not normally distributed and had unequal variances; therefore, they were ranked and compared by the Mann–Whitney U test. The median percent change in CD34+/CD133+ log_10_ cell counts was significantly (*p* = 0.006) higher in the EECP responder group (14.6%, interquartile range (IQR) = 23.7%) than non-responder group (−4.5%, IQR = 24.2%) while the median percent change in CD34+/KDR+ log_10_ cell counts was not different (*p* = 0.558) between the responder (4.1%, IQR = 39.8%) and non-responder (5.0%, IQR = 33.9%) groups (Figure 3). Cardiac rehabilitation patients did not have significantly different (all *p* > 0.80) treatment-associated changes in CD34+/CD133+ or CD34+/KDR+ log_10_ cell counts between responders and non-responders (Figure 3). These results confirmed that, collectively and individually, increased levels of CD34+/CD133+ cells were observed in EECP therapy subjects and support a strong association between improved exercise capacity and increases in this adult stem cell population.

### 2.3. Association of Clinical and Experimental Variables with MACE Hazard

MACE occurred in 17 of 39 (44%) EECP patients and in 5 of 11 (45%) of cardiac rehabilitation patients. Events in EECP patients included one myocardial infarction, hospitalization of six patients for heart failure (with four cardiovascular deaths), angioplasties in four patients (three de novo lesions and one restenosis), placement of stents in five patients, and development of unstable angina in one patient (Figure 1). Events in cardiac rehabilitation patients included two cardiac deaths, one death due to non-cardiac-related causes, two angioplasties, and one hospitalization for heart failure (Figure 1). Survival analysis of the combined cohorts of RA patients who underwent EECP or cardiac rehabilitation therapy revealed that improved exercise capacity was significantly associated with a reduced hazard ratio (HR) of MACE (HR = 0.42, 95% C.I. = 0.18–1.00, *p* = 0.041, Figure 4). However, none of the other clinical variables of changes in DP score, TT, and CCS score had a significant (all *p*-values > 0.14) effect on MACE hazard in the combined cohorts. When the two cohorts were analyzed separately, none of the clinical variables, including improved exercise capacity, were significantly (all *p* > 0.14) associated with MACE hazard in either cohort. Univariate and multivariate Cox regression analyses were performed to assess the effects of baseline clinical characteristics as potential confounders for MACE hazard in the two RA patient cohorts. Variables with univariate *p*-values ≤ 0.2 were entered into a multivariate, one-way Cox regression model, and in no case did a single variable or combination of baseline clinical variables have a significant effect (all *p >* 0.09) on MACE hazard (Appendix A). Cox regression analysis showed that the experimental variables of changes in CD34+/CD133+ and CD34+/KDR levels were not significantly associated with MACE hazard either in the combined or individual EECP and cardiac rehabilitation patient cohorts (all *p >* 0.25).

### 2.4. Clonogenic Potential of EPCs

Endothelial cell colony outgrowth assays were performed on peripheral blood samples from three EECP patients and six healthy, age-matched control subjects to determine if EECP treatment mobilized the release of CD34+/CD133+ EPCs with endothelial cell clonogenic potential into the circulation. The average number of outgrowth colonies obtained from baseline and repeat samples of EECP patients was significantly higher (207 ± 35 colonies/dL vs. 30 ± 12 colonies/dL; *p <* 0.001) than those obtained from healthy controls. Outgrowth colonies obtained from sequential blood samples of two EECP patients who responded to therapy increased significantly (*p =* 0.002) over baseline (78 ± 22 colonies/dL vs. 265 ± 64 colonies/dL), while repeat blood samples from two healthy controls showed no significant (*p* = 0.967) increases in outgrowth colonies (baseline = 24 ± 12 colonies/dL vs. repeat samples = 30 ± 12 colonies/dL, Figure 5). The endothelial cell markers, low-density lipoprotein, and platelet endothelial cell adhesion molecule-1 (PECAM) were present in many cells of outgrowth colonies, consistent with an endothelial cell phenotype, and some colonies developed tubules, suggesting vasculogenic potential (Figure 6).

## 3. Discussion

Several findings emerged from this study that have important implications for the utility of mobilizing adult EPCs by EECP therapy to treat conditions involving vascular damage such as RA. First, even with a small patient cohort, this trial reproduced the clinical benefits of EECP that have been demonstrated for RA patients in large-scale registry trials and supports the hypothesis that mobilization of adult stem cells improves collateral circulation in these patients. Secondly, we show for the first time that objective measures of improved exercise capacity after EECP are associated with increased mobilization of CD34+/CD133+ cells but not CD34+/KDR+ cells. Our results are congruent with those of Kiernan et al. [16], who used a slightly different gating strategy to show that CD34+/CD45_dim_/CD133+ cells but not CD34+/CD45_dim_/KDR+ cells were elevated over the course of EECP treatment. Since these experiments were initiated, it has been reported that CD34+/KDR+ cells are a subset of CD19+ B cells whose role in vasculogenesis is uncertain [26]. Since we did not employ a CD19 marker in our experiments, we were unable to independently confirm these findings. However, our results and those of Kiernan suggest that CD34+/CD133+ cells play a more prominent role in vascular repair than the CD34+/KDR+ population in RA patients. Thirdly, we report for the first time that increased CD34+/CD133+ levels are significant predictors of increased exercise capacity in EECP patients, while changes in exercise capacity were not associated with changes in CD34+/CD133+ levels in cardiac rehabilitation patients. Importantly, we demonstrated that CD34+/CD133+ cells isolated from EECP patients who responded with improved exercise have the potential to differentiate into endothelial cells and that the number of differentiated colonies increased during EECP treatments. Lastly, upon analyzing the combined EECP and cardiac rehabilitation cohorts, we found that increased exercise capacity following therapy was associated with a reduced hazard of MACE. These results demonstrate that interventions that increase exercise capacity, as seen in the trials with intracardiac and intracoronary injections of autologous CD34+ stem cells, have a significant beneficial effect on long-term patient outcomes [3]. The fact that the individual variables of changes in DP score and TT, as objective measures of exercise capacity, did not have a significant association with MACE hazard suggests that stratification which considers both variables, as we have done, is a more robust measure of exercise capacity and a better predictor of MACE hazard. Although important for patients’ quality of life, improvement in the subjective measure of CCS score was not associated with a reduced MACE hazard. Changes in EPC levels did not have a significant effect on MACE hazard in either the EECP or cardiac rehabilitation cohort; however, the fact that they were correlated with increased exercise capacity in the EECP cohort suggests they are important contributors to improved clinical outcomes in these patients. The reason EPC levels alone do not correlate with MACE-free survival could be that some patients have increased EPC levels but not improved exercise capacity due to the different extent of the ischemic burden of the individual patient or other inflammatory factors. The correlation between increased CD34+/CD133+ counts and increased exercise capacity that is observed after EECP, but not cardiac rehabilitation therapy, indicates that EECP mobilizes stem cells independently of any exercise training effect. Since mobilization of these precursors is believed to be stimulated by vascular endothelial growth factor (VEGF) and stromal cell-derived factor-1 (SDF-1), which are upregulated by EECP, our data support a mechanistic link between EPC mobilization and EECP that may promote vascular repair and increased collateral circulation [21,27,28,29,30] as evidenced by treatment-associated increases in CD34+/CD133+ cells and outgrowth colonies derived from them with endothelial cell characteristics.

Recent reports on the role of biomechanical pulsation in endothelial-to-hematopoietic stem cells (HSC) transition provide an attractive mechanistic hypothesis that could explain the difference we observed between EECP and exercise training on CD34+/CD133+ mobilization [31,32,33,34]. EECP induces high laminar sheer stress in the vasculature and thus may be activating mechanosensing pathways to stimulate HSC development. This may explain the lack of correlation between increased EPC levels and increased exercise capacity in cardiac rehabilitation patients. Limitations of our study are that the data are observational, it lacks placebo-controlled patients, and it was conducted at a single site with a small number of patients. It is, therefore, only hypothesis-generating. Nonetheless, we confirm the work of others by showing that EPC levels increase during EECP treatment, and unlike other investigators, we have included pre- and post-treatment exercise data and an extended follow-up of over 48 months to evaluate effects on MACE-free survival.

EECP has been shown to reduce hospitalization of RA patients by 56% and improve the quality of life after treatment [35]. Improving EECP efficacy would further save costs and improve the patient’s quality of life. Intracoronary injection of CD133+ EPCs improved the ejection fraction of RA patients at one year [35], and several double-blind studies have demonstrated that injections of CD34+ stem cells, which include the CD133+ cell subset, improved anginal symptoms and exercise capability [36,37,38,39,40]. It is possible that more non-responders may be converted to responders by the administration of pharmaceuticals that mobilize CD133+ progenitors from the bone marrow, such as G-CSF (granulocyte colony stimulating factor) [41]. Since granulocyte-colony stimulating factor (G-CSF) has been successfully administered for mobilization and large-scale isolation of CD133+ progenitors from healthy adult donors, it may be that G-CSF could improve EECP responses if given during treatment [41]. As the population ages and comorbidity increases, non-invasive treatment options for RA that reduce angina, improve the quality of life, and reduce the cost of caring for these patients, which is estimated to be up to USD 33,000 per year, should be adopted, and improved upon [42]. Our study demonstrates that the mobilization of CD34+/CD133+ stem cells by EECP should be explored as a treatment option.

## 4. Materials and Methods

### 4.1. Subject Characteristics

A total of 51 patients with RA, defined as having Canadian Cardiovascular Society (CCS) Class I to IV angina pectoris pain not relieved with medication or revascularization procedures, were enrolled consecutively in the study. Thirty-nine patients underwent EECP therapy, eleven patients who chose to undergo cardiac rehabilitation instead served as a comparator cohort, and one patient declined either form of treatment. Patient accrual and treatment were conducted in the cardiology practice of Joseph T. Tartaglia, M.D., Westchester County, NY, from May 2007 to March 2013. Blood samples were drawn weekly or, at a minimum, on three different dates from EECP patients during the treatment period to enumerate the circulating total hematopoietic stem cell population defined as CD45+/CD34+ cells and EPC populations defined as CD34+/CD133+ and CD34+/KDR+ cells by flow cytometry. Cardiac rehabilitation patients had blood drawn at the beginning, middle, and end of their treatments for flow cytometric analysis. Blood samples were also obtained for endothelial cell outgrowth assays from 6 healthy, age-matched subjects who constituted a control cohort of individuals without RA. Cell counts were normalized to the number of CD45+ mononuclear cells (MNC) and are expressed as counts/100,000 MNC. Missing flow cytometric data for one or both EPC populations are indicated in Figure 1 and in Appendix A. All patients underwent exercise stress testing 4 weeks or less prior to initiating treatment and 4 weeks or less after treatment except for 2 EECP patients who experienced a cardiac event and could not exercise after starting treatment, 2 patients who did not complete a post-EECP stress test for non-cardiac related reasons, and 1 patient who could not exercise before starting EECP treatment (Appendix A). Medications were not altered during the 7-week course of treatment. Patients were interviewed prior to treatment to assess CCS class, the amount of nitroglycerin use, and the number of angina attacks. Exclusion criteria included unstable angina, severe aortic insufficiency and large thoracic abdominal aortic aneurysms, uncontrolled congestive heart failure or hypertension, severe rapid arrhythmia, severe peripheral vascular disease, severe uncontrolled hypertension defined as systolic pressure > 170 mmHg and/or diastolic pressure > 100 mmHg, and hypertrophic cardiomyopathy. Before each session, patients were interviewed for changes in symptoms, nitroglycerin use, number of angina episodes, and general functional capacity. Potential sources of bias were avoided in the following ways: (1) the data concerning patient angina scores and diastolic augmentation were collected by a certified EECP technologist who had no knowledge of the clinical outcomes or EPC levels and (2) the treadmill data and MACE outcomes were collected in a blind fashion by a physician without knowledge of the EPC levels. Patients were followed for a median of 54 months for MACE to determine if EECP-associated changes in objective measures of exercise capacity (TT and DP scores) or EPC levels influenced MACE-free survival. All patients in the primary practice reached an endpoint of the study or were followed for a pre-planned period of at least four years. Two patients’ records were requested from their physicians with the patient’s permission. Baseline clinical data are recorded in Appendix A with missing values indicated, and the time after treatment of each patient’s last follow-up contact is recorded in Appendix A. 

### 4.2. EECP Treatments

Patients were treated with 1 h of EECP daily 5 days a week for 7 weeks for a total of 35 h of treatment with a model T3 device (Vasomedical Inc., Plainview, NY, USA). The timing of inflation and deflation of pressure cuffs was varied by a physician or a certified technologist who adjusted the pressure of the cuffs from 200 ± 40 mmHg to maximize diastolic augmentation.

### 4.3. Cardiac Rehabilitation Therapy

The 12-week cardiac rehabilitation program included 36 one-hour-long sessions (3×/week) of individualized, ECG telemetry-monitored aerobic, physiologist-guided exercise. During the program, patients were taught to assess their tolerance for exercise by developing self-monitoring skills. They also attended educational seminars on maintaining cardiac well-being and adopting a heart-healthy diet.

### 4.4. Treadmill Exercise Stress Tests

Stress tests were performed according to the Bruce protocol except in two cases where a modified Bruce protocol was necessary. Treadmill times (TT), double product (DP) scores, and the metabolic equivalent of work were recorded pre- and post-EECP. Patients were followed clinically for a median of 54 months (range 5 to 112 months) for MACE, defined as cardiovascular death, any myocardial infarction, coronary angioplasty that revealed restenosis or de novo lesions, coronary artery bypass surgery, unstable angina, and hospitalization for heart failure.

### 4.5. Multicolor Flow Cytometry

Between 3 and 10 mL of peripheral blood were drawn into ethylene diamine tetraacetice acid (EDTA)-coated tubes once a week for each week of the EECP treatment. Samples were processed for flow cytometry and/or stem cell isolation within 3 h of collection. White blood cells recovered after Histopaque (Sigma-Aldrich, Cat# 10771, St. Louis, MO, USA) separation or erythrocyte lysis were washed with PBS/0.1% BSA and incubated with fluorochrome-labeled antibodies to enumerate CD45+/CD34+, CD34+/CD133+, and CD34+/KDR+ cells after morphological gating on 2-dimensional side-scatter and forward-scatter plots to exclude granulocytes. Fluoroscein isothiocyanate (FITC)-labeled CD34 (Cat# 130-081-001, RRID:AB_244350), APC-CD133 (Cat# 130-098-829, RRID:AB_2660883), and Vioblue-CD45 (Cat# 130-113-122, RRID:AB_2725950) antibodies and human erythrocyte lysis kits (Cat# 130-098-456) were purchased from Miltenyi Biotec Inc., San Diego, CA, USA, and phycoerytherin (PE)-labeled KDR (Cat# FAB357P, RRID:AB_357165) antibodies were purchased from R and D Systems, Minneapolis, MN, USA. Flow cytometry experiments were performed on a MACSquant flow cytometer (Miltenyi Biotec, Inc., Bergisch Gladbach, Germany) with appropriate color compensation and analyzed with MACSquantify software (Version 2.6.1515.13751).

### 4.6. In Vitro Clonogenic Potential of EPCs

CD34+/CD133+ cells were isolated from 7 mL of EDTA-treated peripheral blood from a subset of RA patients and six healthy volunteers with a magnetized anti-CD133 microbead and column purification kit (Miltenyi Biotec, CD133 MicroBead kit, Cat# 130-100-830 and MS columns, Cat# 130-042-201) according to the manufacturer’s protocol. The purified CD34+/CD133+ cells were assessed for clonogenic potential after culture for 4 weeks in a modified Hill assay [43,44]. The assay was performed by culturing cells in 4-well plates (Nalge Nunc International, Rochester, NY, USA) containing 500 µL plasmin gels prepared from autologous plasma recovered from the blood sample. Ten percent plasma gels were prepared in Iscove’s medium with thrombin (1 U/mL Sigma-Aldrich, St. Louis, MO, USA). Cells were cultured on solidified gels for 48 h in Iscove’s medium with 20% fetal bovine serum (Sigma), 50 ng/mL basic fibroblast growth factor (FGF, PreproTech, Rocky Hill, NJ, USA), 10 ng/mL VEGF (PreproTech), 10 μL/mL endothelial cell growth factor (acidic FGF plus heparin; Sigma), and penicillin/streptomycin (Sigma). Following the initial 48 h culture period, non-adherent cells were transferred to single 2 cm^2^ wells of plastic plates coated with 50 µg/cm^2^ of fibronectin (Sigma) and cultured for four weeks. Fifty percent of the culture medium was replaced twice weekly with fresh medium, and the total number of outgrowth colonies contained in a well was visually counted after two weeks of incubation. For LDL uptake assays, live cells in plasma gels were incubated for 4 h at 37 °C with sterile, 10 μg/mL Dil-Ac-LDL (Biomedical Technologies Inc.) diluted in culture medium. Following labeling, gels were washed 3× with PBS and fixed with formalin for 20 min in the dark before being imaged, as described by Eisenberg et al. [45]. To visualize CD31-positive cells and outgrowth colonies, plasma gels were fixed with 10% neutral buffered formalin prior to incubation with PECAM (CD31) antibody (Santa Cruz Biotechnology, Inc., Dallas, TX, USA), following the protocol recommended by the manufacturer.

### 4.7. Statistical Analysis

All data were analyzed with Number Cruncher Statistical Software (NCSS 12, version 12.0.16, 2020, Kaysville, UT, USA). Two-sample *t*-tests were applied for two group comparisons of data with normal distributions and equal variances, and the Aspin–Welch *t*-test was applied for data with normal distributions and unequal variances. Data with non-normal distributions and unequal variances were ranked and analyzed with the Mann–Whitney U test for two groups and with the Kruskal–Wallis multiple comparison Z test with Bonferroni correction for pairwise comparisons of more than two groups. The medians and interquartile ranges (IQR) are reported for ranked data. Longitudinal cell count data were converted to log_10_ values to stabilize the variance and meet the assumption of normality and are expressed as log_10_ cell counts ± SEM (Standard Error of the Mean). They were evaluated for significant changes over the treatment period by repeated measures analysis of variance (ANOVA). Logistic regression analysis was performed to assess the predictive value of treatment-related changes in EPC counts on changes in exercise capacity. Kaplan–Meier and Cox regression survival analyses were performed to test the effects of treatments and baseline clinical variables on MACE hazard. Censored data due to loss of follow-up are indicated in the Kaplan–Meier curves and are accounted for in both types of survival analysis. The significance level was set at *p* = 0.05 for all tests. A sample size calculation using the ClinCalc online calculator (https://clincalc.com/) accessed on 29 November 2022 yielded an *n* of 32 patients when set to a desired power of 0.80 with the pre-determined goal of detecting at least a 25% difference in the means of continuous random variables in datasets with estimated coefficients of variation of 0.25.

## Figures and Tables

**Figure 1 ijms-25-10030-f001:**
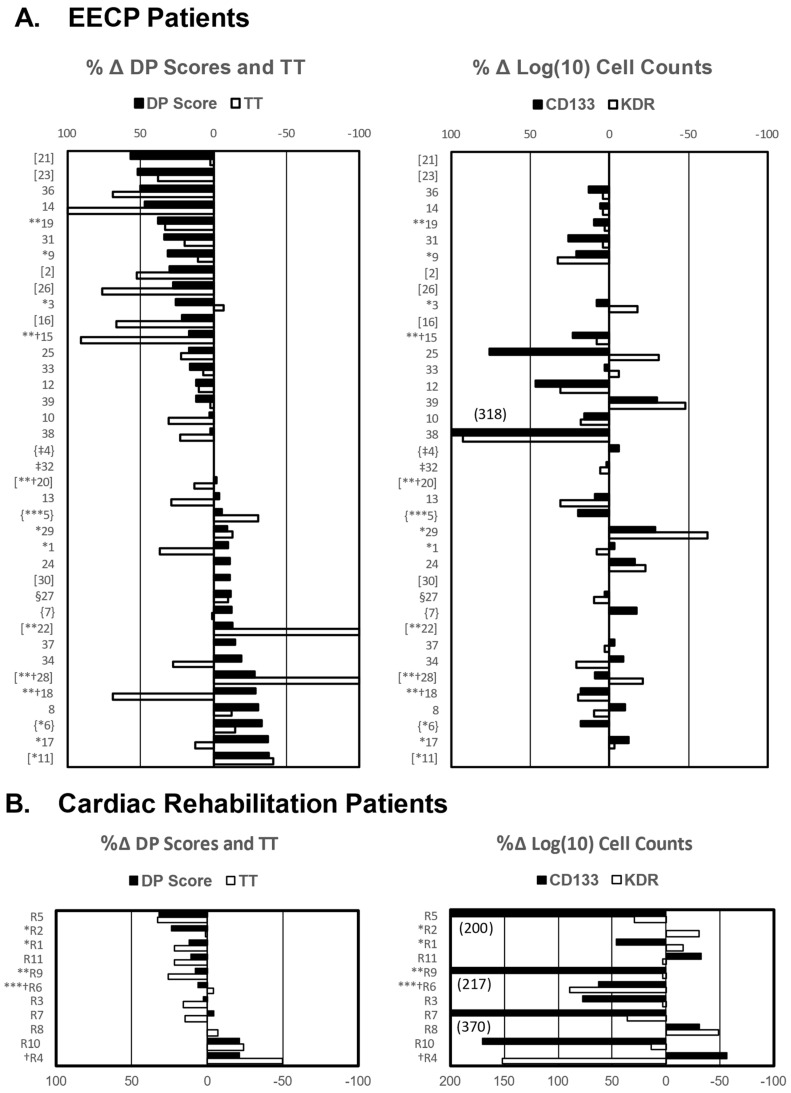
Percent changes in DP scores (solid bars) and TT (open bars) and in log_10_ transformed CD34+/CD133+ counts (solid bars) and CD34+/KDR+ counts (open bars) and occurrence of MACE following EECP and cardiac rehabilitation therapy. (**A**) The percent change (Δ) in stress test measurements ([post-therapy measurement minus pre-therapy measurements]/pre-therapy measurement × 100) and (**B**) the percent change (Δ) in log_10_ transformed cell count measurements ([log_10_ of mean week 4–7 cell count minus log_10_ of mean week 0–3 cell count]/log_10_ of mean week 0–3 cell count × 100) are shown for each patient. In instances where a patient was unable to do a post-therapy stress test, the percent change in TT was recorded as a 100% decrease, and when a patient could not do a pre-therapy stress test but could do a post-therapy test, the percent change was recorded as a 100% increase. The percent change in DP for patients who could not exercise either before or after EECP was calculated as described above based on their resting DP score. Patients who did not complete a post-therapy stress test for non-cardiac related reasons are denoted by (‡) and were not entered in survival analyses. The types of MACE patients experienced are identified as follows: * angioplasty findings of de novo lesion or restenosis; ** heart failure, *** myocardial infarction, and § unstable angina. Deaths are denoted by † (patient R4 had a sudden, non-cardiac-related death). Patients for whom insufficient data were available to calculate treatment-related changes in CD34+/CD133+ and CD34+/KDR+ cell counts are enclosed in brackets [patient #], and those for whom CD34+/KDR+ data were not available are enclosed in braces {patient #}. Percent changes in cell count greater than 200% are indicated in parentheses by the bars (EECP patient *n =* 37, rehabilitation patient *n* = 11).

**Figure 2 ijms-25-10030-f002:**
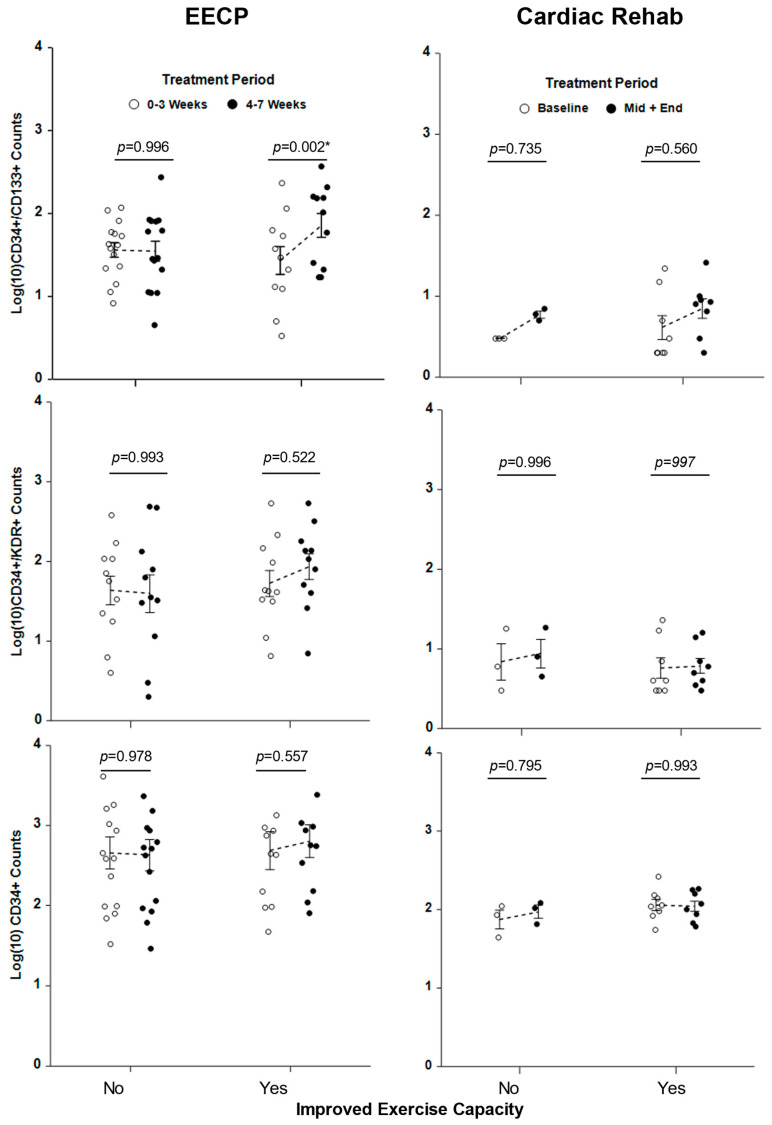
Circulating levels of CD34+/CD133+ increase over the course of EECP therapy in patients who respond with increased exercise capacity. Counts of circulating CD34+/CD133+, CD34+/KDR+, and total CD45+/CD34+ cells from patients undergoing EECP treatment and cardiac rehabilitation therapy are expressed as log_10_ (cell counts/100,000 mononuclear cells) and are plotted against the treatment periods after being stratified into patients who did (Yes) or did not (No) respond to the therapies with improved exercise capacity. The data were analyzed by repeated measures ANOVA and tested for significant differences between cell counts in the first (weeks 0–3, open circles) and the second half (weeks 4–7, closed circles) of EECP therapy and for differences between baseline cell counts (open circles) and those observed in the second half (middle + end, closed circles) of rehabilitation therapy. Error bars represent the standard error of the mean. EECP patient *n* = 26, cardiac rehabilitation patient *n* = 11. * Significant at alpha = 0.05.

**Figure 3 ijms-25-10030-f003:**
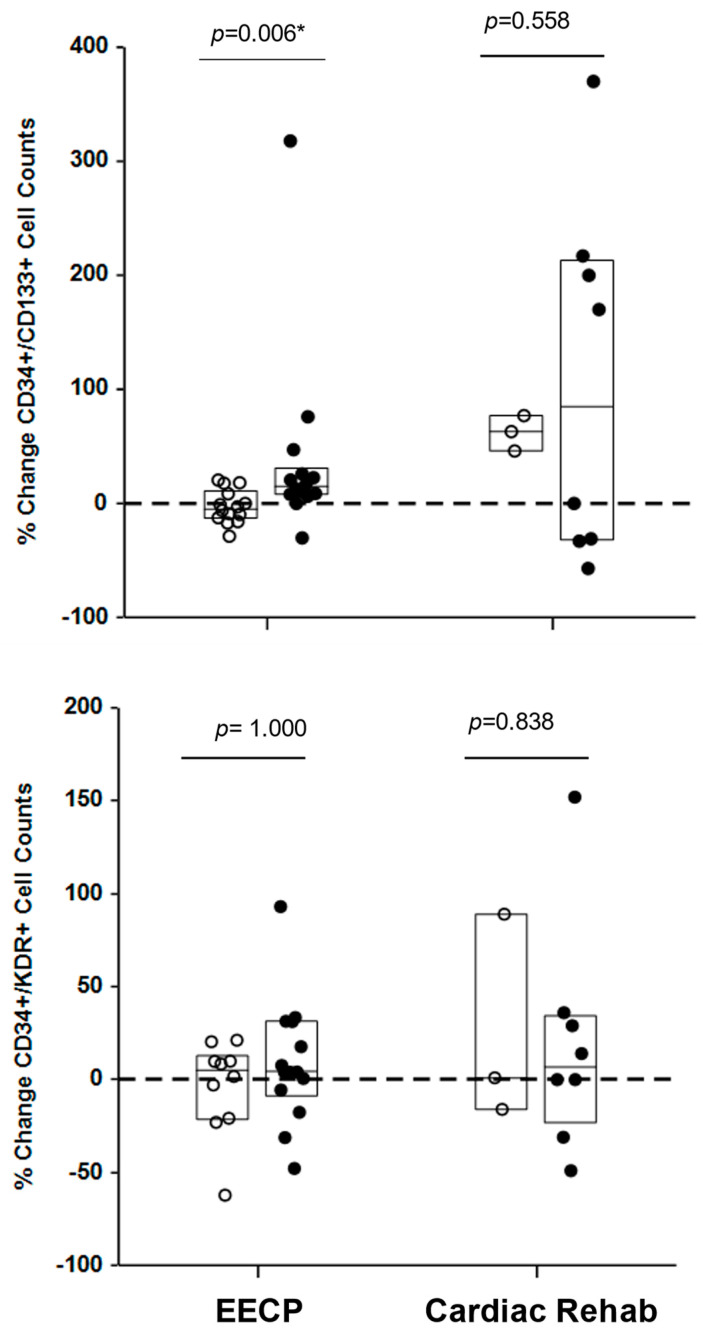
EECP responders had larger treatment-related increases in CD34+/CD133+ cell counts than non-responders. The percent change in CD34+/CD133+ and CD34+/KDR+ cell counts between the first and second half of EECP therapy and over the course of cardiac rehabilitation therapy for individual patients are plotted with the median value and IQR (box plot) for each group. Differences between percent changes in EPCs in responder (closed circles) and non-responder (open circles) groups in each treatment were tested for significance by the Mann–Whitney U test on ranked data since these data did not meet the assumptions of normality or equal variance. EECP patient *n* = 26, cardiac rehabilitation *n* = 11. * Significant at alpha = 0.05.

**Figure 4 ijms-25-10030-f004:**
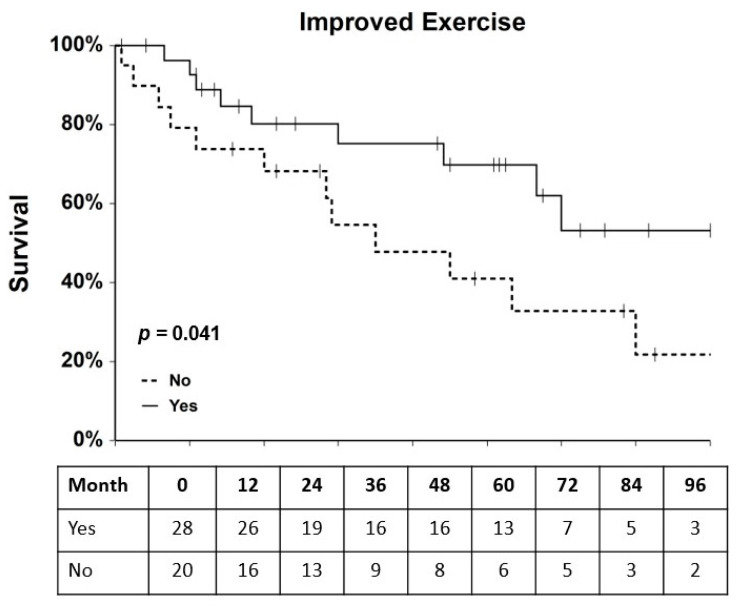
RA patients who responded to EECP and cardiac rehabilitation with improved exercise capacity had a reduced risk of MACE. RA patients were stratified according to four criteria: (1) improved TT or not, (2) improved DP score or not, (3) improved CCS score or not, and (4) responder to therapy with improved exercise capacity or not and evaluated for their effect on MACE hazard by Kaplan–Meier survival analysis of the combined EECP and cardiac cohorts and the cohorts separately. In this analysis, only improved exercise capacity had a significant effect on MACE hazard and only for the combined EECP and rehabilitation patient cohorts. In the survival curves shown for the combined cohorts, responders are represented by a solid line and non-responders by a dashed line. Tick marks on the curves denote times at which patients were lost to follow-up. The numbers of patients in the improved (yes) or not improved (no) categories at each 12-month interval are shown in the table below the survival curves.

**Figure 5 ijms-25-10030-f005:**
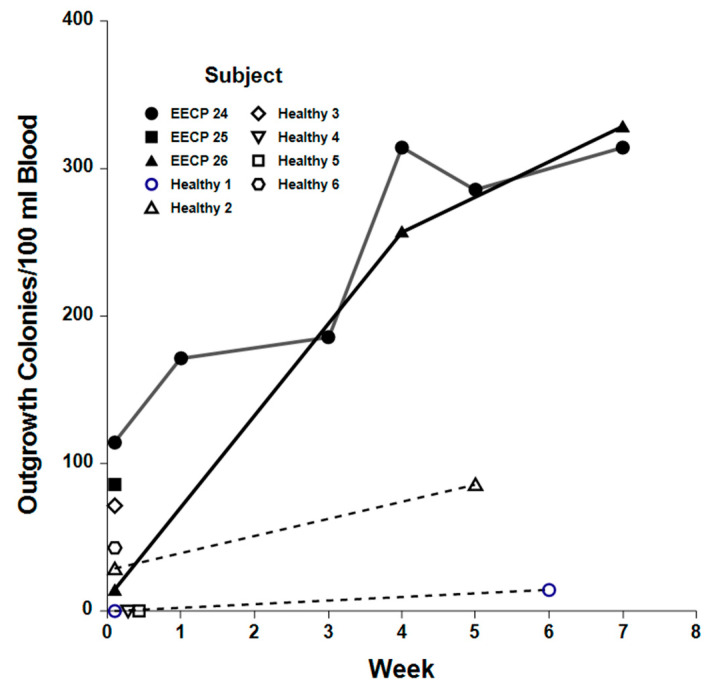
EECP therapy induces increased numbers of EPC outgrowth colonies. Numbers of outgrowth colonies obtained from CD34+/CD133+ cells isolated from EECP-treated RA patients during therapy (closed symbols) were increased (*p =* 0.022) over baseline levels and over those of age-matched healthy volunteers (open symbols, *p <* 0.001). Sequential blood samples taken from EECP-treated RA patients #24 and #26 and healthy control subjects 1 and 2 yielded the number of outgrowth colonies/dL of peripheral blood for the weeks indicated after the baseline samples were taken. The number of outgrowth colonies obtained from single blood samples of EECP-treated RA patient #25 and healthy control subjects 3-6 are denoted by separate symbols at the 0-time point.

**Figure 6 ijms-25-10030-f006:**
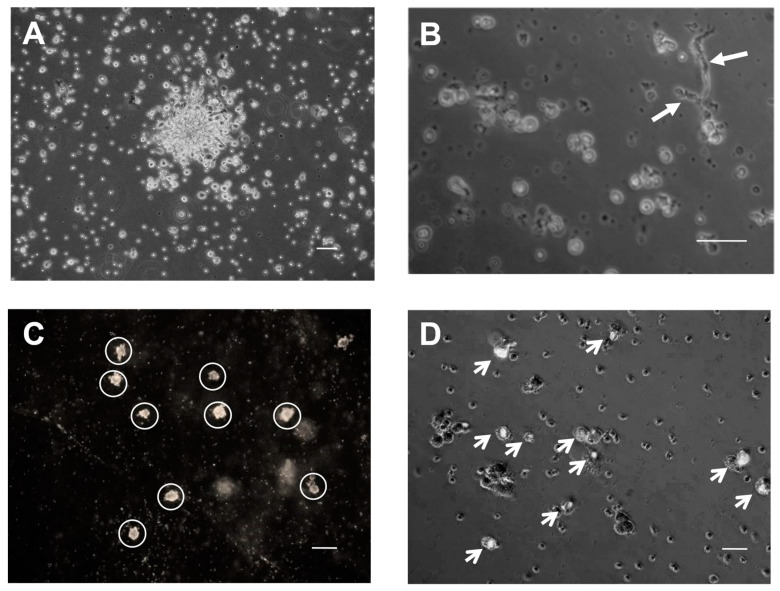
Outgrowth colonies of CD34+/CD133+ cells selected from the peripheral blood of EECP patients show endothelial cell characteristics. CD34+/CD133+ cells selected from 7 mL volumes of blood from EECP patients and healthy volunteers following RBC lysis were cultured in modified Hill clonogenic assays as described in the Methods. The total numbers of outgrowth colonies contained in quadruplicate culture wells were visually counted after 2 weeks of incubation and averaged. Examples of initial outgrowth colonies (panel (**A**)) and the development of a vascular network (arrows, panel (**B**)) are shown. After 4 weeks in culture, cells were stained for PECAM (denoted by circles in panel (**C**)) and low-density lipoprotein (denoted by arrows in panel (**D**)) expression (scale bar = 60 µm).

**Table 1 ijms-25-10030-t001:** Circulating levels of adult stem cell populations in RA patients at the beginning and end of EECP and cardiac rehabilitation therapy.

Therapy	Circulating Cell Population (Counts/100,000 CD45+ Mononuclear Cells)					
	CD45+/CD34+		CD34+/CD133+		CD34+/KDR+	
	Start	Finish	Start	Finish	Start	Finish
EECP	2817 ± 1513	2544 ± 1356	50 ± 9	81 ± 16	98 ± 26	136 ± 33
Cardiac Rehab	113.3 ± 1.7	112.8 ± 13.2	5.6 ± 6.6	8.2 ± 6.4	8.3 ± 2.2	8.1 ± 1.6

## Data Availability

All relevant data are contained within the article. The original contributions presented in the study are included in the article/Appendix A. Further inquiries can be directed to the corresponding author.

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
