# Peer review of "Mobilization of Endogenous CD34+/CD133+ Endothelial Progenitor Cells by Enhanced External Counter Pulsation for Treatment of Refractory Angina"

_ijms, 2024, doi:10.3390/ijms251810030_

Round 1

Reviewer 1 Report

Comments and Suggestions for Authors

In the study, the authors found that mobilization of CD34+/CD133+ by enhanced external counterpulsation is associated with improved exercise
capacity in  patients with refractory angina. The findngs are impressive and practically useful. However, I would like to make some comments.

1. Did the authors FMO standard when cells subtypes are labeled with flow cytometry ?

2. The sentence (lines 99-100) seems to have no logic end.

3.  Did the authors investigate the association between the number of EPCs subsets and CV risk factors, duration of disease, concomitant comorbidties ?

4. Was the functional status of CD34+/CD133+ cells  associated with previous critical conditions, trauma, infections, etc?

Author Response

  1. Did the authors FMO standard when cells subtypes are labeled with flow cytometry ? The staining with the fluorochrome-labeled antibodies selected for these experiments was strong enough to readily discriminate between marker-positive and marker-negative cell populations so fluorescence minus one standardization was not used when enumerating cell subtypes.  
  2. The sentence (lines 99-100) seems to have no logic end. Thanks for catching the truncated sentence.  It should read: "The STROBE 2007 checklist for clinical cohort studies is provided in Appendix A" and has been corrected in the current manuscript.  The checklist was not submitted with the Supplemental Data files since it was not clear when and how it should be added as an Appendix to the submission.  The file will be included as an attachment.
  3.  Did the authors investigate the association between the number of EPCs subsets and CV risk factors, duration of disease, concomitant comorbidties? We did not correlate the level of EPC’s subsets to CV risk factors, duration of disease, and concomitant comorbidities. The patients all had similar baseline characteristics. There was no baseline risk factor, or comorbid conditions that correlated with MACE and we do not think that subset analysis in a small study would likely produce any reliable information. We are not sure what is meant by the duration of the disease, but given that the patients had RA, their coronary vascular disease duration is likely of long-standing. It would be difficult to decide how to measure the beginning of their RA. since a patient's history would be likely unreliable. 
  4.  Was the functional status of CD34+/CD133+ cells  associated with previous critical conditions, trauma, infections, etc? During the 7-week course of treatment of the patients in the study, there were no patients who needed to interrupt the treatments for hospitalizations other than those who reached the endpoint of the study through MACE. We did not track minor infections or trauma, but major trauma such as a motor vehicle accident (i.e. broken bones) would have likely interrupted the treatment, and to our knowledge, no such occurrence happened.

Reviewer 2 Report

Comments and Suggestions for Authors

The authors present an interesting study in which an alternative means for inducing stem cell repair in the body is examined in a cohort f refractory angina patients. Currently, intravenous injection of stem cells is an increasingly popular yet expensive therapy for a variety of physiological complications. EECP therapy, which is FDA-approved, was examined in this clinical population, with results demonstrating EECP therapy increases circulating levels of stem cells with improved exercise capacity in those in receipt of such. Ex vivo examination showed the increase in the stem cells population had the potential to differentiate into endothelial cells, highlighting the therapeutic potential of the approach in contrast to the restrictive alternative mentioned above.

In reviewing the manuscript, I made a couple of observations. The following should be considered by the authors when preparing a suitable revision.   

1.      In the version received it appears as if an author is missing from the list. The list should be reviewed in advance of any resubmission.

2.      The labelling in the graphs is somewhat confusing with the patient identifiers utilising a complex system for denoting each participant. The authors should review their approach to designated patients an identity in the study such that it is clearer to track and interpret the trends on display.  

3.      The n-number should be clearly stated on each graph, or at least in the figure legend that accompanies the figure.

4.      In figure 5, there is reference in the accompanying text that a patient with RA was included in the study yet they are not represented on the graph. Moreover, for this figure, why were only 3 EECP patients included and 6 healthy samples?

5.      There doesn’t seem to be any method for the microscopy analysis of the cells where PECAM and LDL was detected. These details should be included in the methods section.

Author Response

  1. In the version received it appears as if an author is missing from the list. The list should be reviewed in advance of any resubmission. An author is not missing, the "and" was misplaced while the paper was being edited.  The error has been corrected in the current version.
  2. The labelling in the graphs is somewhat confusing with the patient identifiers utilising a complex system for denoting each participant. The authors should review their approach to designated patients an identity in the study such that it is clearer to track and interpret the trends on display.  Patient numbers were assigned based on their date of entry into the study and are displayed in ascending order in the Supplementary Data tables.  We chose to use a waterfall plot in Figure 1 to display the changes in treadmill performance after therapy from largest increase to largest decrease in DP score in order to see how they correlated with changes in patient EPC levels.  The patient #'s were accordingly sorted with their corresponding change in DP score and change in EPC levels.  Also the type of MACE that occurred and EPC data that were available were denoted with each patient #.  The goal was to provide the clearest visual representation of how changes in DP score and TT correlated with changes in EPC levels than would be possible by showing the results in the numeric order of the patients.  A visual inspection of the figure reveals that a larger proportion of patients with improved exercise capacity (as shown by increased DP scores and/or increased TT) have increases in EPC levels than patients whose exercise capacity is not improved.  This correlation was verified by logistic regression analysis which showed that increases in EPC levels were highly significant predictors for improved exercise capacity.  For that reason we would prefer not to change the figure. 
  3. The n-number should be clearly stated on each graph, or at least in the figure legend that accompanies the figure.  The number of EECP and cardiac rehabilitation patients depicted in figures 1-3 are now included in the figure legends. 
  4.  In figure 5, there is reference in the accompanying text that a patient with RA was included in the study yet they are not represented on the graph. Moreover, for this figure, why were only 3 EECP patients included and 6 healthy samples? The text and legend reference to patient #40 with RA who was not treated with either EECP or cardiac rehab should have been deleted since with an n of 1 no meaningful statistical comparison can be made with the other groups.  His inclusion was an oversight in the final editing of the paper and has been corrected in the revised manuscript.  Only a subset of RA patients and a limited number of healthy subjects were selected for this analysis since the clonogenic assays are time consuming and labor intensive.  Even with this relatively small number of subjects there was still sufficient statistical power to detect significant differences in the number of outgrowth colonies between the being and end of EECP treatment and between the number of colonies obtained from healthy age-matched control subjects and EECP patients. 
  5. There doesn’t seem to be any method for the microscopy analysis of the cells where PECAM and LDL was detected. These details should be included in the methods section.  The methodology for PECAM and LDL staining is included in lines 401-406 of the methods section in the revised manuscript and reference #45 has been added for the staining method.